# A close unicellular animal relative and predator of schistosomes exhibits chemokinesis in response to proteins and peptides from its prey

Soniya R. Quick[1], Jason S. Bains[2], Catherine Gerdt[1], Bryan Walker[1], Eleanor B. Goldstone[1], Theresa Jakuszeit[3], Andrew W. Baggaley[2,4], Ottavio A. Croze[2], Joseph P. Gerdt[1]*

1 Department of Chemistry, Indiana University, Bloomington, Indiana, United States of America, 2 School of Mathematics, Statistics and Physics, Newcastle University, Newcastle upon Tyne, United Kingdom, 3 Institut Curie and Institut Pierre Gilles de Gennes, PSL Research University, CNRS UMR 144, Paris, France, 4 Department of Mathematics and Statistics, Lancaster University, Lancaster, United Kingdom

* jpgerdt@iu.edu

## Abstract

Regulated motility is vital for many cells—both for unicellular microbes and for cells within multicellular bodies. Different conditions require different rates and directions of movement. For the microbial predator *Capsaspora owczarzaki*, its motility is likely essential for predation. This organism has been shown to prey on diverse organisms, including the schistosome parasites that co-reside with it in *Biomphalaria glabrata* snails. *Capsaspora* is also one of the closest living unicellular relatives of animals. This phylogenic placement makes *Capsaspora*'s motility an attractive target for understanding the evolution of motility in animal cells. Until now, little was known of how *Capsaspora* regulates its rate and direction of motility. Here we found that it exhibits chemokinesis (increased movement in response to chemical factors) in response to proteins released from prey cells. Chemokinesis also occurs in response to pure proteins—including bovine serum albumin. We found that this chemokinesis behavior is dependent on *Capsaspora* cell density, which suggests that the regulated motility is a cooperative behavior (possibly to improve cooperative feeding). We developed a mathematical model of *Capsaspora* motility and found that chemokinesis can benefit *Capsaspora* predation. In this model, *Capsaspora* moved in random trajectories. Chemotaxis (directional motility along a chemical gradient toward prey) is likely to synergize with this chemokinesis to further improve predation. Finally, we quantitatively analyzed *Capsaspora*'s previously reported chemotaxis behavior. These findings lay a foundation for characterizing the mechanisms of regulated motility in a predator of a human pathogen and a model for the ancestor of animals.

**Data availability statement:** All data are available in IU DataCore (https://doi.org/10.5967/sh2f-xt93).

**Funding:** This work was supported by the National Institutes of Health (R35GM138376) to J.P.G. The content of this paper is solely the responsibility of the authors and does not necessarily represent the official views of the National Institutes of Health. This work was also supported by a Camille Dreyfus Teacher-Scholar Award (TC-24-028) to J.P.G., the Engineering and Physical Sciences Research Council [EP/W524700/1] to J.S.B. and the Human Frontier Science Program (No. LT000941/2021-C) to T.J. In all cases, the funders played no role in the study design, data collection and analysis, decision to publish, or preparation of the manuscript.

**Competing interests:** The authors have declared that no competing interests exist.

## Author summary

How cells decide when to move and how quickly to move is still unclear in many organisms, including in the intriguing protist *Capsaspora owczarzaki*. Because of its close relatedness to animals, *Capsaspora* offers a snapshot into how the unicellular ancestors of animals may have lived. *Capsaspora* has also been found to live in the snails that transmit schistosomiasis, and it can kill the parasitic worms that cause this disease. *Capsaspora* uses thin 'legs' called filopodia to crawl and find prey to consume. This motility may have been important for the ancestor of animals, and it is likely key for *Capsaspora*'s interactions with its host snail and co-resident schistosomes. Here, we discovered that proteins from diverse prey can increase the rate of movement by *Capsaspora* (a process termed chemokinesis). This response depends on the density of *Capsapora* cells, suggesting that the cells may coordinate their movement via communication with each other. We also developed a mathematical model of *Capsaspora* motility, which revealed that chemokinesis is likely to improve the efficiency of *Capsaspora*'s predatory behavior. This finding reveals the significance of chemical signaling for *Capsaspora*'s symbioses and suggests that chemokinesis may have been an important phenotype in the unicellular ancestor of animals.

## Introduction

Motility is a key trait of many cells across the kingdoms of life. Motility allows unicellular organisms to enter environments with more nutrients, evade predators and toxins, and find partners for cooperative behaviors [1,2]. Within multicellular organisms, motile cell types are essential for tissue development, immunity, wound healing, and sexual reproduction [3]. In many cases, motility is tightly regulated. First, cells can regulate *when* they move (and relatedly, how fast they should move). Second, cells can regulate the *direction* in which they move. When the rate of movement is regulated by soluble chemical factors, the process is called chemokinesis; whereas, when the direction of movement is regulated by a gradient of soluble chemical factors, the process is called chemotaxis (Fig 1A) [4]. The molecular mechanisms of chemotaxis and chemokinesis are sparsely elucidated across different lineages, leaving open the question of how evolutionarily conserved these common behaviors are. Furthermore, the ecological implications of chemokinesis are still incompletely understood [5,6].

In order to shed light on the evolution of and ecological benefits of cellular motility, we here study the model protist *Capsaspora owczarzaki* (hereafter *Capsaspora*). *Capsaspora*'s multicellular aggregation behavior has been well studied [7–12]; however, it also migrates in a mesenchymal-like fashion [13] employing its long actin-filled filopodia [14]. This motility behavior is notably less understood than (but just as vital as) the aggregation phenotype. *Capsaspora* was initially isolated from a *Biomphalaria glabrata* snail—a vector that transmits parasitic worms that cause schistosomiasis [15].In that initial report, *Capsaspora* was said to exhibit chemotaxis toward leaking

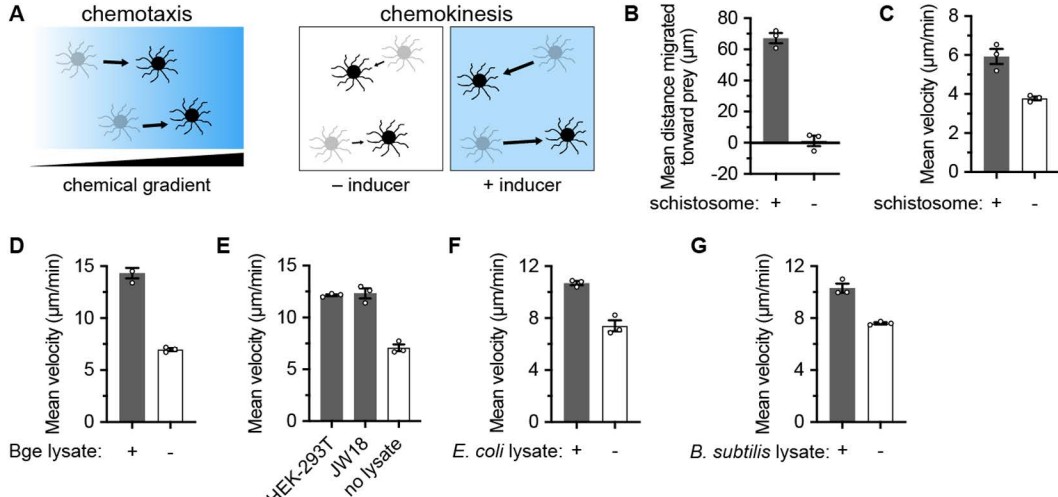

**Fig 1. *Capsaspora* chemotaxis and chemokinesis responses to prey components.** (A) Illustrations of chemotaxis (directional motility along a concentration gradient of a chemoattractant) and chemokinesis (increased rate of cell movement in response to a chemical inducer, without directionality). (B) Chemotaxis of *Capsaspora* toward schistosome sporocysts. Average net movement toward the schistosome is reported (negative value would indicate movement away from schistosome). Data were derived from S1–S3 Videos (with schistosome) and S4–S6 Videos (without schistosome) tracking cells for 4 hours (videos available at https://doi.org/10.5967/sh2f-xt93). To observe maximum movement, only cells that were tracked for ≥ 2 hours were included in the analysis. Only *Capsaspora* cells 70 μm to 240 μm from the schistosome were tracked because cells closer than 70 μm might attach to the schistosome via filopodia, and cells further than 240 μm were near the boundaries of the field of view and may be near other schistosomes outside the field of view. Data for individual cell tracks are reported in S1 Fig. (C) Chemokinesis of *Capsaspora* in response to schistosome sporocysts. Non-directional cell velocity is reported, using cells tracked from the same videos for panel B. Data were only plotted from cells 70–90 μm from the schistosome during the first 15 minutes of each video since they had the maximal chemokinesis response. (D) Chemokinesis in response to lysate from *B. glabrata* embryonic (Bge) cells. Lysate from ~2 × 10⁶ cells was added to 100 μl of *Capsaspora* culture. See S2A Fig for full dose-response data. (G) Chemokinesis in response to lysate from mammalian (HEK-293T) and insect (JW18) cells. Lysates from ~2 × 10⁵ HEK-293T cells and ~4 × 10⁵ JW18 cells were each added to 100 μl of *Capsaspora* culture. See S2B and S2C Fig for full dose-response data. (H) Chemokinesis in response to lysate from *E. coli* cells. Lysate from approximately ~3 × 10⁸ cells was added to 100 μl of *Capsaspora* culture. See S2D Fig for full dose-response data. (I) Chemokinesis in response to lysate from *B. subtilis* cells. Lysate from ~2 × 10⁷ cells was added to 100 μl of *Capsaspora* culture. See S2E Fig for full dose-response data. In all plots, error bars represent standard error of the mean of a biological triplicate of three individual wells of cells (n = 3). Individual biological replicates are displayed with white circles. Within each biological replicate, dozens of individual cells were tracked. Chemokinesis assay method B was used for panels D–G (see Methods). We note that movement appears faster in assay method B compared to the case of schistosome co-culture (panel C). This is believed to be an artifact of 20 × more frequent measurements in method B reporting a truer full distance moved per minute, compared to the one-frame-per-minute displacement values from the co-culture experiment.

schistosome prey. This behavior is proposed to be important for its ability to 'hunt' and devour schistosomes—which could enable *Capsaspora* to clear schistosomes from their snail vectors before they transmit to infect humans. *Capsaspora* is also one of the closest living relatives of multicellular animals. It shares many of the cell signaling and adhesion genes that are key for animal multicellular behaviors [16]. Therefore, it is plausible that *Capsaspora* regulates its motility in ways that are uniquely homologous to animals—which would reveal the evolutionary timeline of cellular migration mechanisms in the animal lineage. Therefore, to determine both the ecological impact of *Capsaspora* motility and the possible ancestry of animal cell motility, we aimed to characterize the regulated motility of *Capsaspora*.

Here, we discovered that *Capsaspora* not only chemotaxes toward leaking schistosomes (as reported before [15]), but it also exhibits increased motility (chemokinesis) in response to leaking schistosomes. Furthermore, this chemokinesis response is promiscuous to lysates of diverse cell types, even including prokaryotes. We found discrete pure proteins that can trigger *Capsaspora* chemokinesis. It is common for cells to desensitize to stimuli over time [17,18], and we found that to be true here with *Capsaspora* chemokinesis, as well. Intriguingly, the chemokinesis response is dependent on the cell density of *Capsaspora*, but not due to direct cell-cell contacts. This may arise from cell-cell signaling through soluble

chemical signals produced by the *Capsaspora* cells. Finally, to determine the benefit of chemokinesis for *Capsaspora*, we modeled its chemokinesis response and assessed if chemokinesis could improve *Capsaspora*'s ability to encounter its prey. This model indeed showed an advantage to chemokinesis, even when the cells moved with random trajectories. Chemokinesis likely also synergizes with chemotaxis to increase the rate of directional motility to further benefit the predator—as has been shown in a bacterial system [5]. In total, this work provides initial mechanistic and ecological insight into regulated motility in a protist that may prevent the spread of schistosomiasis and may reveal the origins of cellular behaviors in multicellular animals.

## Results

### *Capsaspora* exhibits chemotaxis and chemokinesis in response to schistosome sporocysts

The initial report of *Capsaspora*, isolated from *B. glabrata* snails, revealed that *Capsaspora* migrated toward schistosome sporocysts. The sporocysts only attracted *Capsaspora* after they were initially attacked by *Capsaspora* (by random contact) and presumably began to 'leak' their contents into the media surrounding them [15].We repeated this analysis and indeed repeatedly observed migration of *Capsaspora* toward schistosomes that already had other *Capsaspora* cells attached (Figs 1B and S1 and S1–S6 Videos available at https://doi.org/10.5967/sh2f-xt93).

Apart from the direction of movement, we asked if the velocity of *Capsaspora* movement was influenced by the presence of schistosome sporocysts. We discovered that *Capsaspora* showed increased overall cell movement (chemokinesis) when co-cultured with schistosome sporocysts (Fig 1C). Therefore, *Capsaspora* exhibits both chemotaxis and chemokinesis in response to leaking prey. Since chemokinesis is both less studied and easier to monitor than chemotaxis, we aimed to further investigate the chemokinesis behavior in this work.

### *Capsaspora* exhibits chemokinesis in response to cellular components from diverse organisms

We hypothesized that chemical components that leaked from the sporocyst were inducing *Capsaspora* chemokinesis. To test this hypothesis, we assessed if lysate from mechanically lysed prey could increase *Capsaspora* motility. We were unable to obtain enough schistosome sporocysts to test their lysate. However, *Capsaspora* has been shown to adhere and kill *Biomphalaria glabrata* embryonic (Bge) cells [15]—which are readily culturable in large quantities. Therefore, we tested if Bge cell lysate could induce *Capsaspora* chemokinesis. Indeed, we observed increased cell movement in the presence of Bge cell lysate in a dose-dependent manner (Figs 1D and S2A). We further explored if other lysed cells induced chemokinesis. We tested lysates from both a mammalian cell line (HEK293T) and an insect cell line (JW18). Both induced chemokinesis (Figs 1E, S2B and S2C). These data together suggested that *Capsaspora* chemokinesis is induced by general cellular factor(s)—not a highly specific factor produced by only *Capsaspora*'s putative symbionts. To further probe the promiscuity of chemokinesis induction, we asked if even prokaryotic cell lysates could induce *Capsaspora* chemokinesis. We tested lysates of both a gram-negative bacterium (*Escherichia coli*) and a gram-positive bacterium (*Bacillus subtilis*). Both induced *Capsaspora* chemokinesis in a dose-dependent manner (Figs 1F, 1G, S2D and S2E). In sum, *Capsaspora* cell motility increased in response to soluble cell lysates from a remarkably wide range of cells across different domains of life. Therefore, *Capsaspora* chemokinesis is induced by common and/or diverse cellular component(s).

### Bovine serum albumin is sufficient to induce chemokinesis in *Capsaspora*

Given that *Capsaspora* motility increases in response to such a range of lysates, we asked if it responded to an extracellular rich biological substance: fetal bovine serum (FBS). Remarkably, *Capsaspora* increased its movement in the presence of FBS in a dose-dependent manner, as well (Fig 2A). This observation further underscored the breadth of chemokinesis-inducing substances, and it afforded an abundant and stable chemokinesis-inducing material for further mechanistic studies.

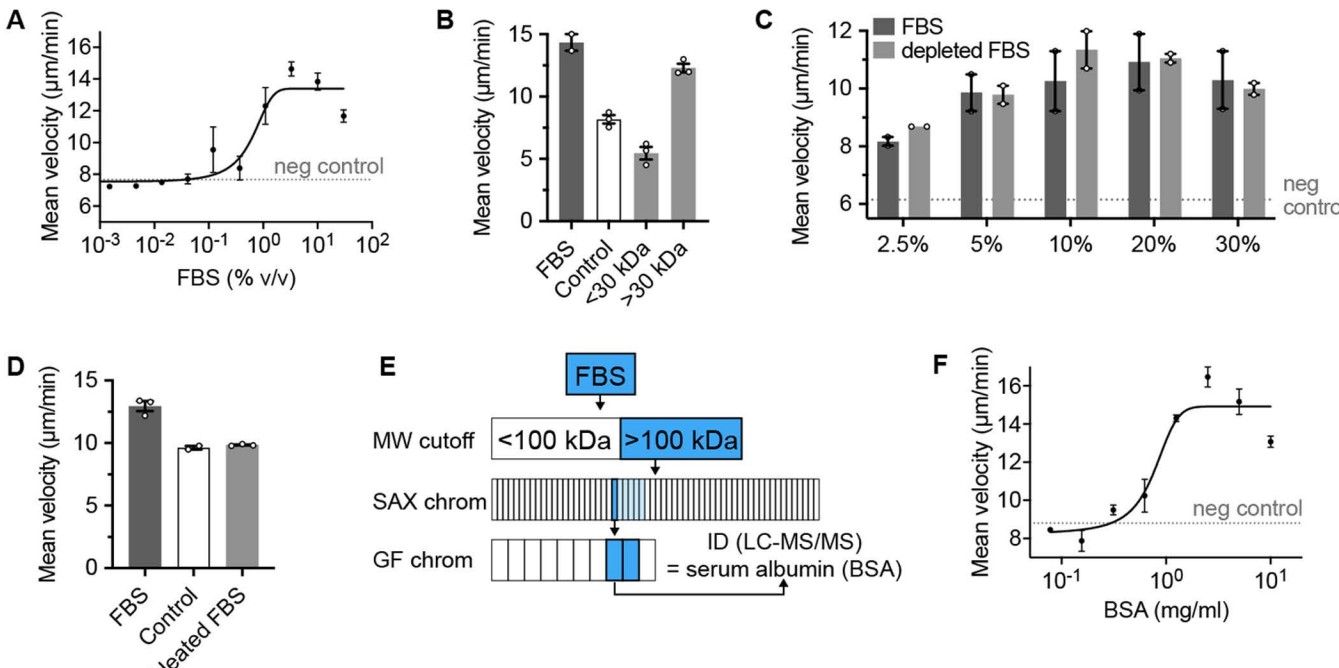

**Fig 2. Bovine serum albumin is sufficient to induce *Capsaspora* chemokinesis.** (A) *Capsaspora* motility upon addition of different concentrations of FBS. The final concentration of FBS in the assay is displayed on the x axis. Error bars represent standard error of the mean of a biological triplicate of three individual wells, each with dozens of cells (n = 3). The dashed line indicates baseline motility upon addition of negative control (water). (B) *Capsaspora* motility upon addition of small molecule (< 30 kDa) or macromolecule (> 30 kDa) components of FBS, relative to full FBS and control (water). Error bars represent standard error of the mean of a biological triplicate of three individual wells of cells (n = 3), except for the FBS positive control, which only had two wells. Individual biological replicates are displayed with white circles. (C) *Capsaspora* motility upon addition of different concentrations of FBS (dark grey) or lipoprotein-deficient FBS (light grey). Error bars represent the range of a biological duplicate of three individual wells, each with dozens of cells (n = 2). Individual biological replicates are displayed with white circles. The dashed line indicates baseline motility upon addition of negative control (Chernin's Balanced Salt Solution, CBSS [20]). (D) *Capsaspora* motility upon addition of heat-treated FBS, compared to untreated FBS and water (control). Error bars represent standard error of the mean of a biological triplicate of three individual wells (n = 3), each containing dozens of cells. Individual biological replicates are displayed with white circles. (E) Bioassay-guided fractionation scheme to purify chemokinesis-inducing substance(s) from FBS. Boxes represent tested fractions from each separation. Dark blue boxes are the most active fractions, which were carried forward. Light blue boxes were less active fractions. Chromatograms, chemokinesis data for individual fractions, and SDS-PAGE image of final active fractions are reported in S3 Fig. LC-MS/MS identification of major proteins is reported in S1 Dataset. (F) *Capsaspora* motility upon addition of different concentrations of BSA. The final concentration of BSA in the assay is displayed on x axis. Error bars represent standard error of the mean of a biological triplicate of three individual wells, each with dozens of cells (n = 3). The dashed line indicates baseline motility upon addition of negative control (water). Chemokinesis assay method A was used for panels A–C, E. Method B was used for panels D and F.

First, we aimed to identify a single pure component from FBS that was sufficient to induce chemokinesis. We investigated if the inducer was a small molecule or a macromolecule by separating FBS with a 30 kilodalton (kDa) molecular weight cutoff filter. We identified that only the > 30 kDa fraction induced chemokinesis—indicating that the inducer is macromolecular (Fig 2B). We previously found that serum lipoproteins trigger cellular aggregation in *Capsaspora* [8,9]. To determine if these lipids were also important for chemokinesis, we tested lipoprotein-deficient FBS, which contains only ~5% of the natural level of lipoproteins. We found that the depleted FBS induced chemokinesis with the same potency as full FBS, indicating that lipoproteins are not necessary for chemokinesis induction by FBS (Fig 2C). We further investigated if the active component was heat-labile. It was, suggesting the active component is a protein or heat-labile glycan (Fig 2D).

We fractionated the macromolecular material from FBS using a strong anion exchange (SAX) column (Figs 2E and S3). We collected one of the most active fractions and subjected it to further fractionation by size exclusion column chromatography (SEC) (Figs 2E and S3). Chemokinesis-inducing activity appeared to correlate with three proteins that migrated at ~70 kDa, ~60 kDa, and ~40 kDa by sodium dodecyl sulfate-polyacrylamide gel electrophoresis (SDS-PAGE) (Figs 2E and S3I). The primary proteins present in these gel bands were bovine serum albumin (BSA), alpha-fetoprotein (AFP), and alpha-2-HS-glycoprotein (fetuin A) (S1 Dataset). These are all major components of fetal serum. BSA is readily available as a purified protein. Therefore, we investigated its ability to induce chemokinesis as a single pure component. We found that BSA could induce chemokinesis in a dose-dependent fashion (Fig 2F). Since FBS contains approximately 25 mg/ml of BSA [19], the $EC_{50}$ value of FBS (~1%, Fig 2A) should contain ~0.25 mg of FBS—which is close to our observed $EC_{50}$ of pure BSA (~1 mg/ml, Fig 2F). Therefore, it is plausible that BSA is the primary (or sole) chemokinesis-inducing agent in FBS. It is also possible that AFP, fetuin A, and/or other proteins may also contribute in proportion to their concentration in FBS, as well.

### *Capsaspora* chemokinesis response is not specific to bovine serum albumin

Since diverse cell lysates (that do not contain BSA) increase *Capsaspora* motility, non-BSA components must also be chemokinesis inducers. Among other readily available proteins, human serum albumin (HSA) and oval albumin also induced migration in *Capsaspora* in a dose-dependent fashion (Fig 3A and 3B). Oval albumin is especially notable, since it is not homologous to BSA. Gamma globulins also appeared to slightly induce motility at only a single concentration (Fig 3C). However, not all proteins increased *Capsaspora* motility—myoglobin failed to induce a significant increase in motility (Fig 3D).

We next considered which moieties of proteins were necessary to promote *Capsaspora* motility. First, the most active proteins are known to be carriers of lipids in the serum. To test if the protein-bound lipids were important for induction, we

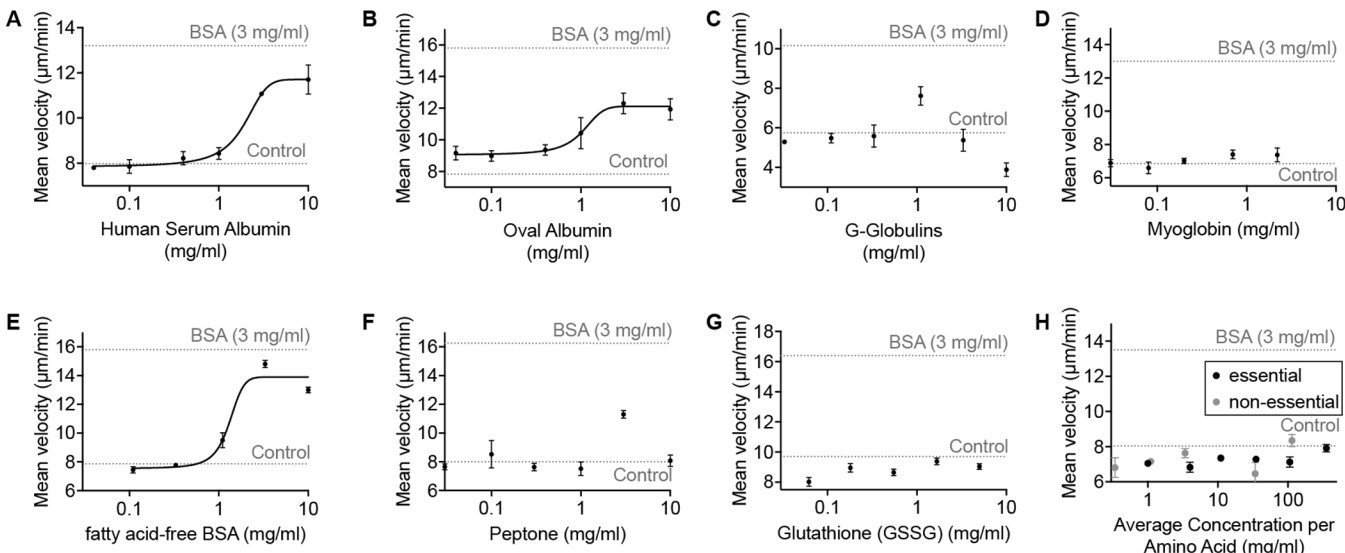

**Fig 3. Other proteins increase *Capsaspora* motility.** *Capsaspora* motility upon addition of different concentrations of (A) human serum albumin (HSA), (B) oval albumin, (C) G-globulins, (D) myoglobin, (E) fatty acid-free BSA, (F) peptone, (G) glutathione, and (H) mixtures of 'essential amino acids' and 'non-essential amino acids'. In all cases, the final concentration of the inducer in the cell culture is displayed on x axis. Error bars represent standard error of the mean of a biological triplicate of three individual wells, each with dozens of cells (n = 3). The dashed line indicates baseline motility upon addition of negative control (water). Chemokinesis assay method B was used for panels A–E, G. Method A was used for panels F and H.

examined chemokinesis by fatty-acid free BSA. This BSA induced motility with a similar potency as native BSA, suggesting that the fatty acids bound to BSA are not important for activity (Fig 3E). Next, we tested if whole protein was necessary or if individual peptides (or even amino acids) could induce chemokinesis. Often simple peptides are ligands for receptors that drive cellular responses. For example, human immune cells respond to N-terminal peptides of bacterial proteins, which contain *N*-formyl methionine [21].Therefore, we tested if individual peptone peptides could induce chemokinesis. They slightly increased motility at a single concentration (Fig 3F). However, another simple peptide (glutathione) failed to induce any aggregation (Fig 3G). Amino acids were incapable of inducing chemokinesis (Fig 3H). These data suggest that some peptides may be sufficient to induce chemokinesis.

## Chemokinesis inducer does not substantially decrease adhesion of cells to surface

To investigate how these inducers increase *Capsaspora* motility, we first asked if BSA influenced the adhesion of *Capsaspora* to surfaces. Surface adhesion is intimately associated with motility. If cells stick too tightly to a surface, they cannot move [22]. Alternatively, if they adhere too weakly, then mesenchymal-type motility is also impossible [23]. Prior work has shown that extensive coating with BSA (for 1 hour) can decrease the adhesion of *Capsaspora* to untreated plastic surfaces [14]. Therefore, we examined if BSA decreased *Capsaspora* adhesion in our chemokinesis experimental conditions (immediate analysis ~2 minutes after addition of BSA to cells adhered to tissue culture-treated surfaces). We found no significant decrease in adherence upon addition of a chemokinesis-inducing concentration of BSA, followed by washing the cells with a level of force that removes approximately half of the surface-attached cells (Fig 4A). Although we cannot rule out a very slight inhibition of attachment speeding up *Capsaspora* movement, our data do not support this hypothesis.

## Incubation with chemokinesis inducer desensitizes *Capsaspora* to further chemokinesis induction

We monitored the kinetics of the increased motility phenotype, and we found that it dissipated over the course of hours (Fig 4B). Initially we suspected that BSA might be depleted from the media, causing the decreased chemokinesis. To test this hypothesis, we monitored the depletion of fluorescently labeled BSA (fl-BSA) from the media. After confirming that the fl-BSA induces chemokinesis (S5A Fig), we collected aliquots of the cell culture media during the chemokinesis experiment conditions. We found that the fl-BSA was not significantly depleted on the timescale of decreased motility (Figs 4C and S5B).

Alternatively, we hypothesized that *Capsaspora* becomes desensitized to its chemokinesis inducer over time. To test this hypothesis, we examined if re-addition of BSA would cause a renewed increase in motility. Even though this re-addition doubled the concentration of BSA, we found no increased motility relative to the untreated control (Fig 4D, black circles vs grey open circles). This desensitization is almost certainly due to the initial BSA treatment, since cells that were not pre-treated with BSA were still capable of a large motility increase upon BSA addition at the later time point (Fig 4D, red "X" marks). In this case the rate of motility decreased more sharply over time—possibly due to the cells running out of nutrients. Ultimately, the chemokinesis induction was temporary and could not be renewed by re-addition of similar concentrations of BSA. This result may indicate an active adaptation of *Capsaspora* to the presence of the chemokinesis inducer, as many organisms adapt to the current concentration of a molecular cue [17,18].

## Chemokinesis response is cell-density-dependent but not contact-dependent

Cell density influences the speed and migration behaviors of different cell types. For example, in multiple cancer cell lines, higher cell density causes faster cell migration compared to lower densities [24,25]. In contrast, some cells in developing animals stop moving when they contact each other at high density [26]. Therefore, we asked if *Capsaspora* cell density regulated its motility in response to BSA. We seeded cells at various cell densities and tested for chemokinesis response to BSA. All previous experiments in this manuscript employed cells at 5,000–10,000 cells per 100 µl volume in a microtiter

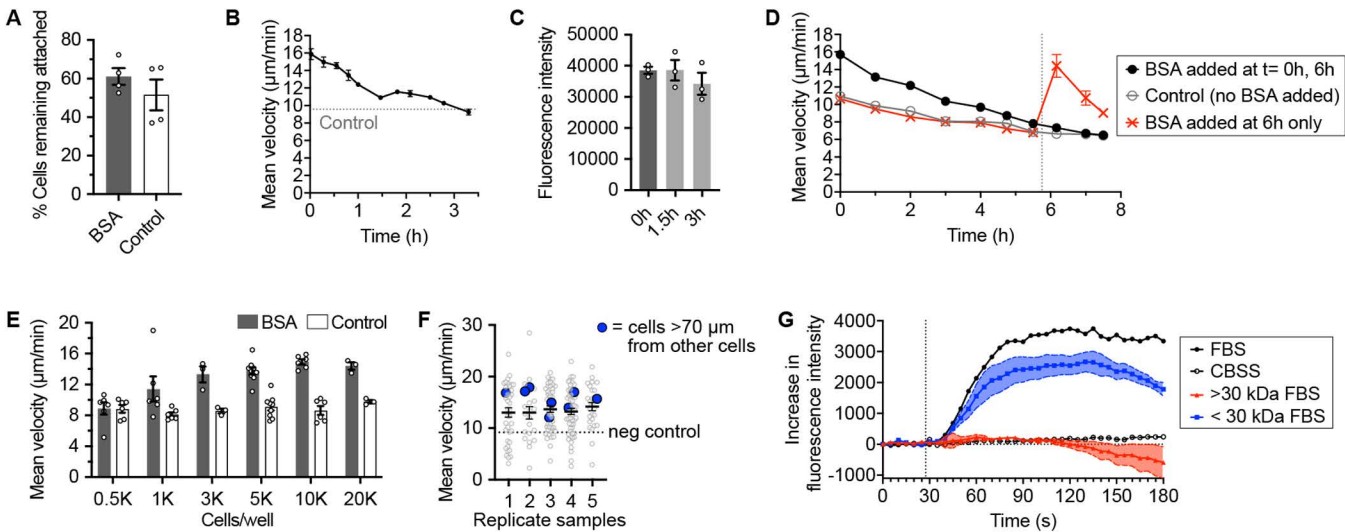

**Fig 4. Insights into chemokinesis induction mechanism.** (A) *Capsaspora* adhesion to tissue culture-treated plastic microplate wells upon addition of 3 mg/ml BSA or control (water) and agitation by pipetting. Error bars represent standard error of the mean of the means of four individual experiments (n = 4). In each experiment, two or three wells were quantified for each condition. The means from each experiment are displayed as white circles. Data from the individual wells from each experiment are in S4 Fig. (B) *Capsaspora* motility at different time points after addition of BSA (3 mg/ml). Error bars represent standard error of the mean of a biological triplicate of three individual wells, each with dozens of cells (n = 3). The dashed line indicates baseline motility before addition of BSA. (C) Quantity of fluorescently labeled BSA remaining in the *Capsaspora* motility assay after 1.5 hours and 3 hours. Error bars represent standard error of the mean of a biological triplicate of three individual wells (n = 3). Individual biological replicates are displayed with white circles. See raw gel image in S5B Fig. (D) *Capsaspora* motility at different time points after addition of BSA (3 mg/ml). Error bars represent standard error of the mean of a biological triplicate of three individual wells, each with dozens of cells (n = 3). The black filled circles represent wells that received BSA (3 mg/ml) at the start and again after ~6 hours. The open grey circles represent wells that received no BSA. The red "X" symbols represent wells that received no initial BSA treatment, but did receive BSA (3 mg/ml) after ~6 hours. The dashed line indicates the time when the ~6 hour BSA dose was added to the black and red conditions. (E) *Capsaspora* motility with different densities of cells upon addition of BSA (3 mg/ml) or control (water). Error bars represent standard error of the mean of two to six replicates of individual wells, each with dozens of cells (n = 3–9). Individual biological replicates are displayed with white circles. (F) Motility of individual *Capsaspora* cells from five replicate wells, highlighting cells that are distant from others (blue, incapable of cell-cell contact) exhibit increased motility on par with the average of their well. Circles represent each tracked cell, line and error bar represent mean and standard error of the mean for that well. Dashed line indicates the mean negative control migration in wells with water added instead of BSA. In all panels, the chemokinesis assays were performed using method B. (G) *Capsaspora* intracellular Ca²⁺ induction (measured via fluorescence of Fluo4-AM dye) with full FBS (n = 1), CBSS negative control (n = 1), and > 30 kDa and <30 kDa FBS components (n = 3 each). Shaded regions indicate standard error of the mean. Vertical dashed line indicates when samples were added to Fluo4-labeled cells.

well. Here we found that *Capsaspora* failed to increase motility at the lowest tested density (500 cells per 100 µl volume in a microtiter well, Fig 4E). Therefore, the chemokinesis response may depend not just on the presence of inducing environmental cues, but also on a *Capsaspora*-derived 'signal'.

This 'signal' may be a soluble factor that is released from cells, or it may be mediated through contacts between adjacent cells [27]. To assess the importance of cell-cell contacts, we examined if cells that were too distant from others for direct contact still exhibited chemokinesis. Prior work has shown that *Capsaspora* filopodia can be up to 24 µm long [14]. We also have not observed filopodia longer than this. Therefore, to conservatively bin cells that should not be contacting other cells, we monitored cells that were > 60 µm away from any other cell in the well. We compared the chemokinesis response of these 'distant' cells to the average of the cells in their well and found that the distant cells migrated as fast as the average (Fig 4F). Therefore, we conclude that the cell-derived chemokinesis signal is not dependent on cell-cell contacts. It may be a secreted soluble molecule.

In sum, it may be that the ultimate chemokinesis 'signal' is a soluble factor, which is produced by *Capsaspora* and secreted in response to the addition of the chemokinesis cue. Alternatively, the density-dependent signal may be

constantly secreted by *Capsaspora* and synergize with the environmental cue to essentially provide an AND logic gate where both the *Capsaspora* signal and the environmental cue must be present for chemokinesis to proceed.

## Chemokinesis is not induced via Ca²⁺ signaling

Chemokinesis and chemotaxis responses are regulated by the $Ca^{2+}$ secondary messengers in cell types including neutrophils and fibroblasts [28,29]. In the protist *Dictyostelium discoideum*, cellular $Ca^{2+}$ levels spike upon addition of the chemoattractant cAMP, but this response does not appear to be necessary for chemotaxis itself [30]. Cellular aggregation in *Capsaspora* also requires the presence of extracellular $Ca^{2+}$, although it is unclear if it is related to signaling [8,12]. To investigate the potential involvement of changes in intracellular $Ca^{2+}$ concentration in *Capsaspora* chemokinesis, we used the cell-permeable $Ca^{2+}$-sensitive dye Fluo4-AM. Although we found that whole FBS induced a stark $Ca^{2+}$-dependent increase in fluorescence, this effect was due to the < 3 kDa small molecules in FBS, not the > 3 kDa macromolecules that induce chemokinesis (Fig 4G). This finding suggests that chemokinesis in *Capsaspora* is not regulated by the $Ca^{2+}$ secondary messenger. Since aggregation is also activated by > 3 kDa lipoprotein components of FBS, this finding also suggests that $Ca^{2+}$ signaling is not involved in inducing *Capsaspora* aggregation. Ongoing work is necessary to further elucidate the non-$Ca^{2+}$-related intracellular signaling mechanisms that regulate both chemokinesis and aggregation in *Capsaspora*.

## Chemokinesis alone can improve predation

Our experiments have revealed that *Capsaspora* cells exhibit chemokinesis in response to various lysates and proteins. While chemotaxis can direct protist predators towards prey [31], the role of chemokinesis in predation is unclear. Previous experiments and mathematical models of predation by bacteria and leukocytes have shown that chemokinesis can improve the efficiency of chemotaxis, but no benefit has been reported for chemokinesis in the absence of chemotaxis [5,6,32,33]. We asked if chemokinesis alone could benefit *Capsaspora*. To explore this, we developed a mathematical model of *Capsaspora* cells responding to a lysed (i.e., leaking) schistosome that exudes a chemokinesis-inducing chemical, see S1 Text.

The theoretical expectation is that organisms displaying chemokinesis in the absence of chemotaxis will accumulate away from the prey (a detrimental behavior for predation) [34]. This expectation stems from the facts that (a) the predator cells will move in random directions in the absence of chemotaxis, and (b) the concentration of the chemo-effector is highest near the source (prey). Therefore, cells near the prey will quickly diffuse randomly away from the prey and then slow down and accumulate far from the prey. However, one feature of *Capsaspora* predation that may mitigate this problem is its irreversible attachment to large prey. Once *Capsaspora* gets close enough to its prey, it will attach to the prey and not diffuse away until the prey is fully devoured—which can take hours (or possibly longer, depending on the size of the prey and number of *Capsaspora* cells attached). Since chemokinesis does not change the predator cells' trajectories, we do not expect it to significantly impact the number of predators that ultimately encounter prey over an infinitely long timescale. However, chemokinesis should increase the rate at which each predator cell randomly encounters the prey and attaches to it. For the cells that do encounter the prey, the decreased time to encounter would increase the amount of time attached to the prey and feeding (i.e., "residence time"). Therefore by this logic, chemokinesis could be advantageous for predation.

To explore this hypothesis, we simulated a population of protists interacting with a schistosome, assumed spherical for simplicity. The schistosome was also assumed stationary, based on the very small movement of these organisms in our in vitro experiments (S1–S6 Videos, available at https://doi.org/10.5967/sh2f-xt93). Further, the schistosome was assumed to exude chemicals with the same diffusivities, and eliciting the same chemokinetic responses for *Capsaspora*, as bovine serum albumin (BSA, Fig 2F). Details of the simulations can be found in S1 Text. Three replicate simulations were run for both a condition with chemokinesis enabled and one without, each over the course of 2 hours (in line with

the length of experiments). As shown in Fig 5A, the simulations predict a higher probability of attachment with chemokinesis enabled during the course of 2 hours. Furthermore, the cells that attach to the prey in the chemokinesis simulation attach earlier than the cells that attach in the non-chemokinesis case (Fig 5B), which gives them an even longer residence time. When considering both the higher number of attached cells and the longer amount of time attached, the 'cumulative residence time' of all cells attached to the prey was ~1.25× higher when chemokinesis occurred (Figs 5C and S6). Overall, our model predicts that chemokinesis alone can be an advantageous predatory behavior when the predator irreversibly attaches to its prey to feed for a long time.

Another way that chemokinesis may aid predation is by increasing dispersion away from recently devoured prey. If the prey releases a large amount of chemokinesis-inducing chemicals, this would increase cell speed, and thus diffusion, encouraging higher dispersal away from the region where there is no longer prey. Thus, randomly moving protist cells need to search for less time before encountering another not-yet-devoured schistosome, which benefits the predator by decreasing time "searching" and increasing the time "residing" attached to prey. To explore this hypothesis, we repeated our simulations, this time using an initial chemical profile that models several recently devoured schistosomes in the vicinity of an intact one. Overall, we did not observe a significant change in the cumulative residence time compared to the standard chemokinesis simulation (Figs 5C and S6). The high diffusivity of the chemo-effector may explain this observation. Since the consumed cells stop secreting the chemo-effector, its concentration dissipates quickly relative to the rate of cell movement.

In sum, our model predicts that chemokinesis alone (without chemotaxis) can be beneficial for predation when the predator irreversibly attaches to its prey. Chemokinesis significantly increased attachment rates of predators to prey over a 2 hour simulation. Our simulation did not include chemotaxis (i.e., directional bias in motility toward prey), and we observed some variation in cumulative residence times in all conditions across repeated simulations (Figs Ib and Ic in S1 Text). Therefore, since random motion has a large effect in predation success, it seems likely that chemotaxis, acting in concert with chemokinesis, can provide an even greater advantage over chemokinesis alone. Specifically, chemotaxis

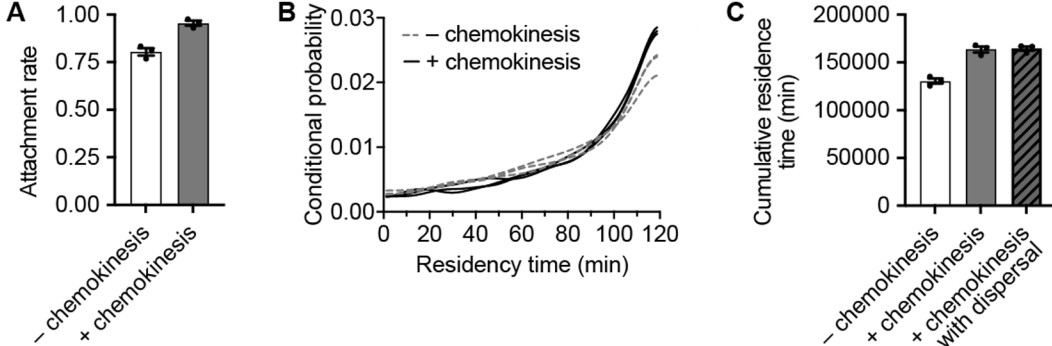

**Fig 5. Numerical simulations of predator-prey interaction reveal benefit for chemokinesis.** (A) Attachment rates (i.e., the fraction of *Capsaspora* cells that attached to the schistosome prey) during a 120-minute simulation of *Capsaspora* cells predating a schistosome with or without chemokinesis to a chemo-effector (with the properties of BSA) exuding from the schistosome. When the cells reach the schistosome, they irreversibly attach to it. (B) Probability of different residence times over the 120 minute simulation period for cells that ultimately attached to the schistosome. "Conditional probability" is the probability of a cell attaching to the schistosome at that moment, given the condition that the cell attaches at some point during the 120 minutes. Cells that never attached are not included in the plot. Three replicate simulations for each condition are shown as individual curves. Curves were generated from kernel density estimates. (C) Cumulative residence times (i.e., the total sum of time that the 2000 *Capsaspora* cells are attached to the schistosome over the 120 minute simulation) with and without chemokinesis. Additionally, the diagonally shaded bar represents the case where chemokinesis is enabled *and* four nearby schistosomes were depleted and gave a final burst of chemo-effector at the start of the simulation—no significant difference was observed compared to the standard chemokinesis condition. See S6 Fig for the full kinetic analysis. In all bar plots for this figure, error bars represent standard error of the mean of a triplicate of three individual simulations, each with 2000 cells (n=3). Individual simulation means are represented with dots.

would bias motion of predators towards beneficial trajectories, reducing randomness and increasing success in predation, as mathematical models have predicted in the case of bacteria [5]. To show this, the chemoattractants to which *Capsaspora* responds would need to be identified and quantified, which is beyond the scope of this study. This would allow our mathematical model to be extended, parameterized, and compared with experimental results.

## Discussion

Here, we have reported that *Capsaspora* exhibits chemokinesis in response to damaged schistosomes, to cell lysate from diverse organisms, and to bovine serum albumin (BSA). This discovery complements earlier work revealing *Capsaspora*'s ability to sense and respond to chemical factors from its snail host and schistosome prey. Namely, *Capsaspora* forms large cellular aggregates in response to lipids from its host snail [8–10], and it chemotaxes toward yet-unknown chemical cues from damaged schistosome prey (initially observed by Owczarzak et al. [15] and quantified here in our work). Therefore, an expanding array of chemical factors from its host and prey shape the behavior of this microbial predator.

We were initially surprised that motility would increase in response to a wide diversity of cues from many sources, including both eukaryotes and bacteria. The response may have evolved to a specific cellular protein that is broadly shared (or has key features that are shared) across domains of life. It may have also evolved to generic abundant features that are shared among many proteins. This generality could actually benefit *Capsaspora* if its predatory behavior is much more promiscuous beyond snails and schistosomes. Although *Capsaspora* has not yet been shown to prey on bacteria, its filasterean relative *Ministeria vibrans* does [35].Therefore it is plausible that a recent ancestor of *Capsaspora* ate bacteria and/or that *Capsaspora* feeds on much more diverse prey than previously thought. Responding to general cues can be advantageous to cells that promiscuously target diverse prey. For example, neutrophils have evolved to sense *N*-formylmethionyl peptides, which are universal in nascent bacterial proteins [21]. Therefore, *Capsaspora*, which actually can prey on many cell types [15], may benefit from sensing common and/or diverse prey cues. Beyond sensing microbial prey, *Capsaspora* may even sense related cues within the snail host, possibly helping it to even discriminate between different tissue environments. In this scenario, there could be multiple 'competing' chemical gradients released from the host and co-resident microbes, which may necessitate more nuanced sensing mechanisms by *Capsaspora*.

We were also surprised by the relatively high concentration of chemokinesis inducer required for maximally increasing motility (~3 mg/ml of BSA). Since the total protein content of cells is typically ~200–300 mg/ml [36], nearby *Capsaspora* can certainly be exposed to prey-derived protein concentrations in excess of 3 mg/ml upon prey lysis. We speculate that this high concentration of protein could have influences beyond a classical ligand-receptor interaction with the *Capsaspora* cells. For example, the proteins may coat the surface of the substrate (or even surfaces of *Capsaspora* cells) to block ligand-receptor interactions, as well. Since surface adhesion is intimately related to motility rates, non-specific protein binding that improves or inhibits cell-substrate adhesion could play a role in the observed motility increases [22,23,37]. We observed no significant influences on cell-substrate adhesion upon brief exposure to BSA; however, minor changes in adhesion that are below our limit of detection may be significant for motility rates. Further work is warranted to explore the role of cell-substrate adhesion in the increased motility of *Capsaspora* near prey. This work would ideally incorporate substrates that are relevant to the natural life of *Capsaspora* (beyond the tissue culture-treated surfaces and fibronectin-coated surfaces [S1B and S1C Fig] that we used in this work).

The observed cell-density dependence of the chemokinesis response is reminiscent of cell-cell communication behaviors during chemotaxis in neutrophils and *Dictyostelium discoideum* [38,39]. In these cases, chemotaxing cells secrete their own signal molecules to relay the message to neighboring cells. In both cases, the ultimate outcome is a cooperative behavior. For neutrophils, this cooperation brings more immune cells to the site of an infection. For *D. discoideum*, a multicellular fruiting body is formed. Likewise, predation in *Capsaspora* is a cooperative behavior, where multiple cells can lyse larger prey and share the released nutrients more efficiently when they co-localize [9,15]. In other words, it could benefit an attached feeding *Capsaspora* cell to attract other cells to help release shared nutrients from the prey. Therefore, it is

sensible for cell-cell signaling to contribute to *Capsaspora*'s regulation of predation. Ongoing work is aimed at characterizing the identity of the putative secreted signal and response mechanism.

The elucidation of the signaling pathways that regulate chemokinesis (and possibly cell-cell communication) in *Capsaspora* could provide information about the evolution of animals. Cellular motility and cell-cell signaling are essential behaviors for multicellular organisms [3,40], and it is unclear how these behaviors were co-opted by animals from their unicellular ancestors [40,41]. *Capsaspora* is a leading model to reveal mechanisms of multicellular behaviors that directly preceded the origin of animals [42], therefore elucidating its mechanisms of cell-cell signaling and regulated motility may reveal how these behaviors were regulated in the ancestors of animals. Our limited investigation in this paper only suggests a lack of involvement of calcium ion signaling (in contrast to many animal cells that leverage $Ca^{2+}$ signaling to regulate motility) [28,29,43]. Therefore, we hypothesize that other secondary messengers are key. Their elucidation will provide points of conversation and divergence with regulation of motility and cell-cell communication in extant animals, providing insight into the origins of these essential multicellular behaviors in the animal lineage.

Finally, chemokinesis is a mysterious phenotype. In contrast to chemotaxis, it does not obviously lead to improved predation—although many predators and symbionts do employ it [6,32,44,45]. On its own, a gradient of a chemokinesis-inducing agent leaking from prey may actually be detrimental to predation—causing dispersal of cells in regions further away that induce lower motility [37]. However, in our model of *Capsaspora* motility, we found that chemokinesis alone was actually able to significantly improve *Capsaspora* predation. We believe the key difference is that *Capsaspora* irreversibly attaches to its prey for a long period of feeding. This long-term attachment upon encountering prey provides a benefit to quick attachment, enabling the significant advantage to chemokinesis (in the absence of chemotaxis). It is likely that chemokinesis could be an even more beneficial predatory trait when it synergizes with chemotaxis, which can bias cell movement towards prey, reducing randomness [5]. Thus the ability of *Capsaspora* to chemotax may compound with the benefit of chemokinesis. Given our demonstration of chemokinetic responses of *Capsaspora* to bacterial lysates, in future studies it would be interesting to apply our simulation framework to explore the role of chemokinesis in the predation of bacteria by protists that do not obviously make use of chemotaxis, such as *Acanthamoeba* [46].

## Conclusion

In sum, we have found that a predatory microbe's motility is increased by proteins released from leaking prey. The behavior is density-dependent, suggesting cooperative predatory behavior. While this increased motility near prey was not initially expected to benefit the predator in the absence of chemotaxis, our simulations suggested that it actually can be beneficial on its own. However, the combined effect of chemotaxis and chemokinesis is likely to be even more beneficial for predation. Given that the studied predator (*Capsaspora owczarzaki*) is one of the closest living relatives of animals, further dissection of the mechanism of chemokinesis induction may inform the evolutionary pathway of regulated cell motility in animals. Notably, *Capsaspora* chemokinesis did not appear to be regulated by calcium ion signaling, unlike motility regulation in many animal cells.

## Methods

### Materials used

Reagents used in this study are summarized in Table 1.

### Cell strain and growth conditions

*Capsaspora owczarzaki* cell cultures (strain American Type Culture Collection ATCC 30864) were grown axenically in 25-cm$^2$ culture flasks with 6 ml ATCC media 1034 (modified PYNFH medium) containing 10% v/v heat-inactivated Fetal Bovine Serum (FBS, Corning 35–011-CV), hereafter growth media, in a 23 °C incubator. Cells were obtained during

**Table 1. Reagents used in this study.**

| Reagent | Cat. No. # | Company |
|---|---|---|
| TC-treated 96-well plate | CLS3997 | Corning |
| TC-treated 96-well plate | 353219 | Corning |
| FBS | 35-011-CV | Corning |
| Lipoprotein-deficient FBS | S5394 | Sigma |
| Ethanolamine | E9508-1L | Sigma |
| Bovine serum albumin | A9418-5G | Sigma |
| Human serum albumin | A1653-1G | Sigma |
| Oval albumin | A5503-1G | Sigma |
| G-globulins | G7516-1G | Sigma |
| Myoglobin | M0630-250MG | Sigma |
| Fatty acid-free BSA | 126609-5G | Sigma |
| Poly-L-Lysine | P4832 | Sigma |
| Glutathione | AAJ6216614 | Fisher |
| Peptone | DF0118-17-0 | Fisher |
| Essential amino acid mix | 11-130-051 | Fisher |
| Non-essential amino acid mix | 11140050 | ThermoFisher |
| Fluorescent BSA | A9771-100MG | Sigma |
| Fibronectin | F1141 | Sigma |
| Fluo4-AM dye | F14201 | Invitrogen |
| Pluronic F-127 surfactant | P3000MP | Fisher |

exponential growth phase by passaging ~0.9 – 1.2 ml of adherent cells scraped into suspension within the specified density range, into 5ml of growth media. The culture was incubated at 23 °C for 24 h to achieve desired density range of $3 \times 10^6$ to $20 \times 10^6$ cells/ml.

### *S. mansoni* miracidia isolation and transformation into primary sporocysts

*S. mansoni* miracidia were isolated according to published protocols [47,48]. Briefly, 3 infected mice livers were obtained from the Biomedical Research Institute (Rockville, MD) (BRI) Schistosomiasis Resource Center. They were shipped overnight with cold packs and added to ice-cold 1.2% sterile NaCl solution immediately after arrival using a sterile 50 ml centrifuge tube. The tubes with the livers were shaken vigorously, and any floating debris and the saline solution was aspirated. The livers were then poured into the ice-cold cup of a Waring blender with another 30 ml of ice-cold sterile 1.2% NaCl solution and blended for 15–25 seconds non-stop making sure the livers were well-blended. The blended livers were centrifuged at 800 × g for 10 min at 4 °C. The supernatant was discarded, and the pellets were washed with ice-cold sterile 1.25% NaCl solution. The pellets were then resuspended with 40 ml room temperature sterile artificial pond water (recipe from BRI, 0.46 µM $FeCl_3$, 220 µM $CaCl_2$, 100 µM $MgSO_4$, 310 µM $KH_2PO_4$, 14 µM $(NH_4)_2SO_4$ in water adjusted to pH 7.2 with NaOH) and transferred into the aluminum-covered sterile volumetric flask. The flask was carefully filled with sterile pond water to approximately 3 cm above the foil covering the neck. The flask was covered with a small petri dish and a bright light shone across the top of the flask. The eggs started to hatch at room temperature. When miracidia had amassed at the surface after 10 min, around 10 ml of the pondwater with miracidia was transferred to a sterile petri dish. The collected miracidia were centrifuged at 290 × g at 4 °C for 2 minutes. Only 1 ml of pondwater was left with the miracidia pellet and another 1ml of Chernin's Balanced Salt Solution (CBSS+) [20] was added to the petri dish. The miracidia were then transferred to a 24-well plate and incubated at 26 °C overnight to stimulate miracidial transformation into primary sporocysts.

## Chemotaxis/chemokinesis near schistosomes

*Capsaspora* cells were counted and resuspended such that the final cell density was $60 \times 10^3$ cells/ml. 200 µl of the *Capsaspora* cells were seeded in each well of a TC-treated 96-well plate (Corning #353219) and incubated overnight at 26 °C. In the experiment run on fibronectin-coated plated (S1B and S1C Fig), a TC-treated 96-well plate (Corning #353219) was pre-coated with 100 µl of 70 µg/ml fibronectin (Sigma #F1141) in 25% FBS-free media before adding *Capsaspora*. The next day, schistosome sporocysts were centrifuged for 30 seconds at $300 \times g$ at room temperature and the supernatant was aspirated immediately. The pellet was then resuspended with 650 µl 25% FBS-free media. Media was aspirated out of the *Capsaspora* wells and replaced with 200 µl of the sporocyst cell culture to afford a few schistosomes per well. Negative control wells received media without schistosomes. For imaging each well, a field of view was set such that there is at least one sporocyst surrounded by many *Capsaspora* cells. Images were collected every one minute for 4 hours using an A1 Nikon inverted microscope. Cells were then analyzed using Imaris image analysis software (Bitplane, version 10.0.1). ImageProcess was used to invert the image, the schistosome was set as the origin reference, and spots were created. Tracking was enabled. Estimated cell diameter was set to 6.43 µm. Background was subtracted, and the automatic threshold was used to quality filter the spots. Cells 70–240 µm from the origin reference frame were selected because cells closer than 70 µm might attach to the schistosome via filopodia, and cells further than 240 µm were near the boundaries of the field of view and may be near other schistosomes. The autoregressive motion algorithm was used for tracking. To remove spurious readings by floating cells (which appear to migrate abnormally quickly and/or appear/disappear quickly), we excluded cells that migrated > 11.2 µm in adjacent frames or lost tracking for > 3 frames. The 'fill gap' function was disabled, and only tracks ≥3 frames were analyzed. For chemotaxis, the distance moved relative to the reference (schistosome) was reported for each cell track. To observe maximal movement, only tracks that had ≥ 2 hours (120 frames) were included for plotting. For chemokinesis, the distance moved frame-to-frame for each cell in each frame was reported. Only data for the first 15 minutes of the video were plotted, and only for cells 70–90 µm from the schistosome, since this was the timeframe and distance with the most notable chemokinesis.

## Chemokinesis assay

This assay was always performed with the strain ATCC 30864 unless mentioned otherwise. The standard chemokinesis assay was performed on TC-treated 96-well plate (Corning #CLS3997). Either $5 \times 10^3$ cells or $10 \times 10^3$ cells were seeded in 90 µl of FBS-free growth media per well in the TC-treated 96-well plate and allowed to settle for 2 h. This low cell density was chosen to avoid confounding results from cellular aggregation (which is known to occur with *Capsaspora* in the presence of FBS at higher cell densities [e.g., $8 \times 10^5$ cells per well] in low-attachment [non TC-treated plates]) [8–10].

**Method A.**  After seeding cells and settling for 2 h, 85–90% of the FBS-free media was removed by aspiration and replaced with 85 µl Chernin's balanced salt solution (CBSS+). Samples were then added at 10 µl volume such that the total volume in each well was 100 µl. At concentrations tested higher than 10%, less CBSS+ was added, and a larger sample volume was added to reach the desired concentration with a final volume of 100 µl. Chemokinesis activity was observed right after the sample addition by taking bright field images every 5 seconds for 3 minutes using an Agilent BioTek Cytation 10 confocal imaging reader.

**Method B.**  After seeding cells and settling for 2 h, samples were directly added at 10 µl volume such that the total volume in a well is 100 µl. Chemokinesis activity was observed right after the sample addition by taking bright field images every 5 seconds for 3 minutes using an Agilent BioTek Cytation 10 confocal imaging reader.

## Chemokinesis quantification

When the Cytation 10 was used to record cell movement, analysis was performed using BioTek Gen 5 software. Briefly, images were pre-processed using a dark background with background flattening using a rolling ball diameter of 9 µm and image smoothing strength of 2 cycles. Kinetic frame alignment was employed based on the previous valid image with a

maximum offset of 200 μm. The cellular analysis module was employed to track cells. A primary mask was applied using a threshold value of 5000 with a dark background, split touching objects, track objects, and fill holes in masks. Object size was filtered to be 1–100 μm. To remove spurious readings by floating cells (which appear to migrate abnormally quickly and/or appear/disappear quickly), we made a subpopulation of cells that migrated ≤60 μm and were tracked for ≥ 18 frames. The 'average velocity' of this subpopulation of cells was reported as the 'mean velocity'.

## Cell lysis

*Biomphalaria glabrata* embryonic (Bge) cells [49,50] were grown in Bge medium with 10% FBS at 26 °C. The culture was resuspended to $20 \times 10^3$ cells/ml and snap-frozen in 20 μl aliquots until further use. The frozen aliquot was freeze-thawed 5 times with a 20 second thawing period between each cycle. An additional 80 μl of FBS-free media was added to make the final volume 100 μl. The lysed culture was centrifuged at 15,000×g for 5 minutes and the supernatant was collected for further testing.

Both animal cell cultures were freshly received and resuspended to appropriate densities using FBS-free media. The cultures were then centrifuged at 15,000×g for 5 min and the cell pellets were lysed using freeze-thaw method for 5 times with 20 second thawing period between each cycle. The lysed cultures were centrifuged at 15,000×g for 5 min to collect the lysate supernatant.

*E. coli* MG1655 culture was grown overnight in lysogeny broth (LB) at 37 °C with shaking and sub-cultured to 50 ml with 1:50 dilution of the overnight culture into fresh media. The 50 ml culture was incubated an additional 18 hours with shaking at 37 °C. The culture was centrifuged at 10,000×g for 15 min. The cell pellet was resuspended in 1 ml of FBS-free media. The culture was lysed using a probe sonication for 10 seconds followed by a 20-second paused, repeated for a total of 5 min. The lysed culture was centrifuged at 10,000×g for another 15 min and the lysate supernatant was collected for further analysis.

*B. subtilis* 168 culture was grown overnight in lysogeny broth (LB) at 37 °C with shaking and sub-cultured to 50 ml with 1:50 dilution of the overnight culture into fresh media. The 50 ml culture was incubated an additional 18 hours with shaking at 37 °C. The 50 ml culture was centrifuged at 10,000×g for 15 min. The cell pellet was resuspended in 1 ml of 4 mg/ml lysozyme and incubated for 15 min. The culture was centrifuged again at 15,000×g for 15 min and the cell pellet was resuspended in 1 ml of FBS-free media. The culture was lysed using sonication for 10 seconds followed by a 20-second paused, repeated for a total of 5 min. The lysed culture was centrifuged at 10,000×g for another 15 min and the lysate supernatant was collected for further analysis.

## FBS heat-treatment

100 μl of FBS in an Eppendorf tube was heated on dry heat block at 75 °C for 30 minutes. The treated and non-treated FBS was tested for chemokinesis assay. The FBS-free growth media was used as negative control.

## Bioassay-guided fractionation

FBS was initially fractionated using either Amicon Ultra 30 kDa or 100 kDa cutoff filters (Sigma, #UFC5030, #UFC5100) according to the manufacturer's direction. Briefly, the filters were first washed with 500 μl of CBSS by centrifugation at 14,000×g for 15 min. Then 500 μl of FBS was added to the cutoff filter and centrifuged at 14,000×g for another 15 min. The small fractions (< 30 kDa, < 100 kDa) were collected for testing, and 20 μl of each large fraction (> 30 kDa or > 100 kDa) was added to 180 μl of CBSS for testing. 10 μl of each stored fraction was used for testing chemokinesis.

## Isolation using strong anion exchange column (SAX)

**Chromatography sample preparation.** 20 ml of FBS was fractionated using an Amicon Ultra 100 kDa cutoff filter (Sigma, #UFC8100) according to the manufacturer's direction. Briefly, the filters were first washed with 5 ml CBSS+ by centrifuging at 4,000×g for 25 min. Then 10 ml of FBS was added to each cutoff filter and centrifuged at 4,000×g for

50 min. The > 100 kDa was washed by adding 5 ml of buffer A (50 mM Tris Ethanolamine, pH 9.5), centrifuged at 4,000 × g for 25 min, and the concentrated sample was collected. The > 100 kDa fraction was then sterile-filtered using Costar Spin-X Centrifuge filters (Corning, #29442–752).

**Chromatography method.** A GE HealthCare HiPrep Q XL 16/10 column (Fisher, #45-002-068) was used. For the mobile phase, 50 mM Tris Ethanolamine, pH 9.5 was used as buffer A and 50 mM Tris Ethanolamine supplemented with 1 M NaCl, pH 9.5 was used as buffer B with a flowrate of 5 ml/min. The column was first washed with buffer B for 5 min and equilibrated with buffer A using 5 column volumes (CV). The 5 ml sample loop was then washed with buffer A followed by injecting 1.2 ml of the sample (> 100 kDa). Unbound material was washed from the column using 10 CV buffer A. The fractions were then eluted by using a linear gradient from 0–55% B using 30 CV and 55–100% B using 20 CV followed by holding 100% B for 10 CV. All fractions including the flow through were collected using a fixed volume of 5 ml per fraction.

**Chemokinesis sample preparation.** The unbound fraction was pooled and labelled as fraction 1. To simplify, a systematic pooling strategy was employed. Initially, a primary pool was made by combining 50 μl from each of the 30 consecutive eluted fractions and labelled them sequentially from fraction 2 through fraction 9. Each of these pooled fractions were concentrated using the 100 kDa cut off filters, centrifuged at 14,000 × g for 15 min at 10 °C. > 100 kDa fractions were then washed by adding 300 μl of CBSS and centrifuged again at 14,000 × g for 15 min at 10 °C and collected the > 100 kDa fractions. 20 μl of each of these fractions were added to another 70 μl of CBSS to such that the total volume was 90 μl and stored at 4 °C until testing.

Once the active pooled fractions were identified (pools 3 and 4), secondary pools were made by combining 100 μl from 10 consecutive initial fractions and labeled them sequentially. For example, fraction 31 contained original fractions 31–40, while fraction 33 contained fractions from 51–60. The same approach was applied to the other active initial pool 4. Pool 4 was subdivided into fractions 41, 42, and 43, where fraction 42 contained original fractions 71–80. The combined fractions were concentrated and washed as above. After collecting the > 100 kDa fractions, 20 μl of each of these fractions were added to 180 μl of CBSS to such that the total volume was 200 μl. These were stored at 4 °C until testing.

Upon identifying activity in specific secondary pools (pools 33 and 41), a tertiary pooling step was performed. Each active secondary pool was further divided by combining 50 μl from 5 consecutive initial fractions into a new pool and labeled accordingly. For example, fraction 33 was divided into 331 and 332, where fraction 332 contained original fractions 56–60 and fraction 421 contained original fractions 71–75. This iterative process allowed for the stepwise narrowing of active fractions. 20 μl of each of these fractions were added to another 80 μl of CBSS such that the total volume was 100 μl. These were stored at 4 °C until testing.

10 μl from each of these pools were added to the chemokinesis assay and CBSS was used as negative control

## Isolation using gel filtration (size exclusion chromatography [SEC])

**Chromatography sample preparation.** 2 ml of each fraction contributing to the 331 pool (i.e., 51–55) was combined to achieve a final volume of 10 ml to prepare for SEC. First, the 100 kDa cutoff filter was washed with 5 ml CBSS+ by centrifuging at 4,000 × g for 25 min. The sample was then centrifuged at 4,000 × g for 35 min using 100 kDa cutoff filter. To the > 100 kDa fraction, another 3 ml of CBSS was added to the sample and centrifuged for another 30 min using 100 kDa cutoff filter and collected the final > 100 kDa for SEC fractionation.

**Chromatography method.** A HiLoad 16/600 Superdex 200 pg (GE healthcare #28989335) column was used. For the mobile phase, 50 mM Tris Ethanolamine supplemented with 1 M NaCl, pH 9.5 was used with a flowrate of 1 ml/min. The column was first washed with the buffer for 5 min and equilibrated with the buffer using 0.2 column volumes (CV). The 1 ml sample loop was then washed with the buffer followed by injecting 1 ml of the sample (> 100 kDa). The fractions were then eluted by using isocratic condition using 5 CV. All fractions were collected using a fixed volume of 3 ml per fraction.

**Chemokinesis sample preparation.** 500 µl from each fraction containing proteins (5–12) was individually centrifuged at 4,000 × g for 15 min at 10 °C using 100 kDa cutoff filters. The > 100 kDa fraction was washed once with 200 µl CBSS and the remained 20 µl of >100 kDa fraction was directly tested for chemokinesis assay and stored at 4 °C for further analysis.

### Different cell density seeding

After measuring the cell density in a culture flask, the culture was appropriately diluted such that seeding with 90 µl of each diluted culture achieves the desired density.

### Desensitization experiments

Chemokinesis assay was performed using the method B and Gen5 for imaging analysis. 10 µl of 30 mg/ml BSA was added to induce chemokinesis as mentioned at t = 0 and 6h.

### Adhesion assay

$5 \times 10^3$ cells were seeded in 90 µl of FBS-free growth media per well in the TC-treated 96-well plate and allowed to settle for 2 h. After seeding cells, 85–90% of FBS-free media was removed by aspiration and replaced with Chernin's balanced salt solution (CBSS+). A Cytation 10 imaging reader was set to capture ~80% of the well's area to count this initial cell number. After imaging, 10 µl of either BSA or water were added to each well and allowed to sit for 2 min (to mimic the chemokinesis conditions). 85–90% of FBS-free media was removed by aspiration and replaced with 85 µl CBSS+ solution. The Cytation 10 imaging reader was again used to capture ~80% of the well's area to count the remaining cell number. The remaining cell number was divided by the initial cell number to determine the % cells remaining for each well.

### Calcium signaling assay

$7 \times 10^4$ cells were seeded in 200 µl of FBS-free growth media per well in the TC-treated black-walled 96-well plate (Corning 353219) and allowed to settle for 2 h at room temperature. After seeding cells, 85–90% of FBS-free media was removed by aspiration and replaced with 100 µl Chernin's balanced salt solution (CBSS+) containing 5 µM Fluo-4 AM dye with 0.1% DMSO and 0.02% pluruonic F-127 surfactant. The cells were incubated in the dark for 45–60 minutes before aspirating the solution and replacing with 100 µl of fresh CBSS +. Cells were imaged using a Olympus IX83 P2ZF inverted microscope fitted with a Yokogawa CSU-W1 spinning disk confocal system, a Hamamatsu ORCA-Flash4.0 camera, and a UPlanSApo 20x/0.75 objective. Fluorescence excitation was performed with a 488 nm laser, and emission was detected at 519 nm with 2x2 binning. Images were taken every 5 seconds for 3 minutes. After 25 seconds of baseline readings, 10 µl of test sample was added (i.e., FBS, CBSS, > 30 kDa FBS, < 30 kDa FBS). Images were analyzed using Imaris image analysis software (Bitplane, version 10.0.1). ImageProcess was used to invert the image, and spots were created. Tracking was enabled. Estimated cell diameter was set to 3.00 µm. Background was subtracted, and the automatic threshold was used to quality filter the spots. The Brownian motion algorithm was used for tracking. To remove spurious readings by floating cells (which appear to migrate abnormally quickly and/or appear/disappear quickly), we excluded cells that migrated > 2.30 µm in adjacent frames or lost tracking for > 3 frames. The 'fill gap' function was disabled. The center intensity for each spot (i.e., each cell) was exported for each frame along the entire track. In Microsoft Excel, cell tracks without all frames included were removed. For each cell track, the initial spot intensity was subtracted from all spot intensity readings, thereby normalizing the initial intensity for every cell to zero. For each well at each time point, the normalized spot intensities for each cell were averaged (n ~ 200 cells). Three replicate wells were analyzed, and their mean ± standard error were plotted (n = 3).

### Numerical modelling of predator prey interaction with chemokinesis

Simulation of a population of *Capsaspora* cells was completed in MATLAB; details can be found in S1 Text. *Capsaspora* cells were modelled as individual agents, using Monte-Carlo methods to account for stochasticity. Rules of motion were inferred using a data driven approach. We analyzed the motility statistics of WT *Capsaspora* cells on a uniform substrate from Video 1 in reference [14]. We found that *Capsaspora* cells display strong directional persistence following a truncated normal distribution for the correlation between direction of currently extended filopodia (Fig C in S1 Text) and the direction of the next extended filopodia. This implies that cells tend to carry on travelling in the same direction with only minor adjustments to the right or left, making larger adjustments rarely, so they would take a long time to reverse direction. We then measured distance travelled after extension of filopodia and found it follows an exponential distribution (Fig B in S1 Text), so that distances travelled following filopodia extensions are fairly uniform. To model chemokinesis, we parameterized the curves found for BSA as well as HSA, which were found to have the same shape with the only difference being BSA saturates at a higher locomotion speed, details can be found in S1 Text.

The schistosome was then modelled as a stationary circle, that would cause cells to become irreversibly attached if they made their way to the schistosome surface. The schistosome would then exude the chemokinesis-inducing chemical at a rate dependent on the local attachment rate of *Capsaspora* cells to the surface. Dispersion of chemical was modelled in a continuum as standard isotropic diffusion with a constant diffusivity. These simulations were run with and without chemokinesis to determine if chemokinesis was beneficial to predation by increasing residency time of *Capsaspora* at the schistosome surface. The simulations were then run again for the case of a chemical profile reflecting several recently lysed schistosomes, to determine if they would act as a 'dispersal agent' and encourage cells to find new prey.

### Supporting information

**S1 Fig. *Capsaspora* chemotaxis to schistosome prey.** (A) Migration of individual cells toward schistosome prey (same data that is summarized in Fig 1B). The three left samples have a schistosome present. The three right samples do not ('prey' location was a randomly chosen spot in the middle of the field of view). Lines indicate means, and error bars represent standard error of the mean. Each circle is a single tracked cell. One-way ANOVA with multiple comparisons was performed, showing that all samples with a schistosome had significantly higher motility toward the schistosome than all three samples lacking a true schistosome. (B–C) Repeated chemotaxis experiment using fibronectin-coated dishes instead of plain tissue culture-treated surfaces (parallel to Figs 1B and S1A). (B) Chemotaxis of *Capsaspora* toward schistosome sporocysts. Average net movement toward the schistosome is reported (negative value would indicate movement away from schistosome). Error bars represent standard error of the mean of a biological triplicate of three individual wells of cells (n = 3). Individual biological replicates are displayed with white circles. Within each biological replicate, many individual cells were tracked. (C) Migration of individual cells toward schistosome prey (same data that is summarized in S1B Fig). The three left samples have a schistosome present. The three right samples do not ('prey' location was a randomly chosen spot in the middle of the field of view). Lines indicate means, and error bars represent standard error of the mean. Each circle is a single tracked cell. One-way ANOVA with multiple comparisons was performed, showing that all samples with a schistosome had significantly higher motility toward the schistosome than all three samples lacking a true schistosome.
(TIF)

**S2 Fig. *Capsaspora* chemokinesis is induced by cell lysates in a dose-dependent manner.** Data match those plotted in Fig 1C–1G, where the data from only the most active concentration was plotted for each lysate. In all cases except for *E. coli* lysate, this was the highest concentration tested. (A) *Capsaspora* motility upon addition of different concentrations of Bge cell lysate. (B) *Capsaspora* motility upon addition of different concentrations of HEK-293T cell lysate. (C) *Capsaspora* motility upon addition of different concentrations of JW18 cell lysate. (D) *Capsaspora* motility upon addition of

different concentrations of *E. coli* cell lysate. (E) *Capsaspora* motility upon addition of different concentrations of *B. subtilis* cell lysate. In all cases, the final concentration of lysed cell material added in the cell culture is displayed on the x-axis. Error bars represent standard error of the mean of a biological triplicate of three individual wells, each with dozens of cells (n = 3). The dashed lines indicate baseline motility upon addition of negative control (water) or induced motility upon addition of 3 mg/ml BSA.
(TIF)

**S3 Fig. Bioassay-guided fractionation of FBS.** (A) Chromatogram of FBS separated by anion exchange (AEX) chromatography. Coarse pooled fractions are shown. (B) *Capsaspora* motility upon addition of coarse fractions from AEX. (C) Chromatogram of FBS separated by anion exchange (AEX) chromatography. Medium-sized and fine-sized pooled fractions are shown. (D) *Capsaspora* motility upon addition of medium-sized pooled fractions from AEX. (E) *Capsaspora* motility upon addition of fine-sized pooled fractions from AEX. (F) Chromatogram of FBS separated by size exclusion chromatography (SEC). Fractions collected are shown. (G) *Capsaspora* motility upon addition of fractions 5 and 6 from SEC. (H) *Capsaspora* motility upon addition of fractions 7–10 from SEC. (I) Coomassie-stained SDS-PAGE gel of SEC fractions. Boxes are placed around gel bands that were excised for identification by tryptic peptide LC-MS/MS. For all motility assays, positive control (10 µl whole FBS) and negative control (10 µl CBSS) were included on the same day as the co-plotted sampled. In all plots, error bars represent standard error of the mean of a biological triplicate (in a couple cases duplicate) of individual wells, each with dozens of cells (n = 3). Individual replicates are displayed with white circles.
(TIF)

**S4 Fig. Replicate experiments testing impact of BSA on *Capsaspora* surface adhesion.** *Capsaspora* adhesion to tissue culture-treated plastic microplate wells upon addition of 3 mg/ml FBS. Each panel is an experiment run on a separate day. The mean from each experiment was used to generate Fig 4A. Error bars represent standard error of the mean of two or three individual wells, each containing dozens of cells. Individual biological replicates are displayed with white circles.
(TIF)

**S5 Fig. Chemokinesis induction and lack of depletion of fluorescent BSA.** (A) *Capsaspora* motility upon addition of 3 mg/ml of BSA or fluorescently modified BSA (fl-BSA), compared to a negative control (water). Error bars represent standard error of the mean of a biological triplicate of three individual wells, each with dozens of cells (n = 3). Individual biological replicates are displayed with white circles. (B) Fluorescence-scanned SDS-PAGE gel used to quantify the remaining fluorescent BSA after incubation with *Capsaspora*. The fl-BSA bands were quantified to generate Fig 4C.
(TIF)

**S6 Fig. Chemokinesis increases cumulative residence time of predators feeding on prey.** Cumulative residence times (i.e., the total sum of time that the 2000 *Capsaspora* cells are attached to the schistosome over the 120 minute simulation) with and without chemokinesis over time. Three replicate simulations for each condition are shown as individual curves. Grey dashed curves show the non-chemokinesis condition. Black solid curves show the chemokinesis condition. Red solid curves represents the case where chemokinesis is enabled *and* four nearby schistosomes were depleted and gave a final burst of chemo-effector at the start of the simulation—no significant difference was observed compared to the standard chemokinesis condition. Fig 5C summarizes the cumulative residence times at the endpoint (120 minutes).
(TIF)

**S1 Dataset. Tryptic peptide LC-MS/MS results for identifying proteins in chemokinesis-inducing fractions of FBS.** (Tab 1) Summary of identified proteins in all three excised bands. (Tabs 2–4) Proteins identified in each gel band 1–3, ranked by relative intensity.
(XLSX)

**S1–S3 Videos. Movement of *Capsaspora* near a schistosome.** Three biological replicate videos tracking the movement of *Capsaspora* cells near a schistosome. Each video was tracking different *Capsaspora* cells around a different schistosome in a different microtiter plate well. One image taken per minute. Available at IU DataCore ([https://doi.org/10.5967/sh2f-xt93](https://doi.org/10.5967/sh2f-xt93)).
(ZIP)

**S4–S6 Videos. Movement of *Capsaspora* without schistosomes.** Three biological replicate videos tracking the movement of *Capsaspora* cells without schistosomes added. Each video was tracking different *Capsaspora* cells in a different microtiter plate well. One image taken per minute. Available at IU DataCore ([https://doi.org/10.5967/sh2f-xt93](https://doi.org/10.5967/sh2f-xt93)).
(ZIP)

**S1 Text. Details of numerical model of chemokinesis.** Document contains extensive details on the development of the chemokinesis model.
(PDF)

**S1 APCK. Chemokinesis model code.** Folder contains code for running the chemokinesis model.
(ZIP)

## Acknowledgments

We thank the Schistosomiasis Resource Center for provision of schistosomes and Bge cells, which were provided by the Schistosomiasis Resource Center of the Biomedical Research Institute (Rockville, MD). NIH: We thank Margaret Mentink-Kane and André Miller for instruction on schistosome isolation. We thank the Light Microscopy Center at Indiana University for support in image acquisition and analysis. We also thank the Indiana University Laboratory for Biological Mass Spectrometry and Jon Trinidad for proteomics assistance. We thank Emily Layton and Richard Hardy for providing mammalian and insect cell pellets. We also thank the entire Gerdt lab for insights and support that helped advance this project.

## Author contributions

**Conceptualization:** Soniya R. Quick, Jason S. Bains, Bryan Walker, Joseph P. Gerdt.

**Data curation:** Jason S. Bains.

**Formal analysis:** Soniya R. Quick, Jason S. Bains, Catherine Gerdt, Joseph P. Gerdt.

**Funding acquisition:** Joseph P. Gerdt.

**Investigation:** Soniya R. Quick, Jason S. Bains, Catherine Gerdt, Bryan Walker, Eleanor B. Goldstone.

**Methodology:** Soniya R. Quick, Catherine Gerdt, Bryan Walker, Eleanor B. Goldstone, Theresa Jakuszeit.

**Project administration:** Joseph P. Gerdt.

**Resources:** Theresa Jakuszeit.

**Supervision:** Andrew W. Baggaley, Ottavio A. Croze, Joseph P. Gerdt.

**Validation:** Soniya R. Quick, Andrew W. Baggaley, Ottavio A. Croze.

**Visualization:** Soniya R. Quick, Jason S. Bains, Joseph P. Gerdt.

**Writing – original draft:** Soniya R. Quick, Jason S. Bains, Joseph P. Gerdt.

**Writing – review & editing:** Soniya R. Quick, Jason S. Bains, Theresa Jakuszeit, Andrew W. Baggaley, Ottavio A. Croze, Joseph P. Gerdt.

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
