## [Decision Letter · Decision Letter 0]

29 Jun 2025

PPATHOGENS-D-25-01241

A close unicellular animal relative and predator of schistosomes exhibits chemokinesis in response to proteins and peptides from its prey

PLOS Pathogens

Dear Dr. Gerdt,

Thank you for submitting your manuscript to PLOS Pathogens. After careful consideration, we feel that it has merit but does not fully meet PLOS Pathogens's publication criteria as it currently stands. Therefore, we invite you to submit a revised version of the manuscript that addresses the points raised during the review process.

Please submit your revised manuscript within 60 days Aug 28 2025 11:59PM. If you will need more time than this to complete your revisions, please reply to this message or contact the journal office at plospathogens@plos.org. Please include the following items when submitting your revised manuscript:

We look forward to receiving your revised manuscript.

Kind regards,

Dominique Soldati-Favre

Section Editor

PLOS Pathogens

 Sumita Bhaduri-McIntosh

Editor-in-Chief

PLOS Pathogens

orcid.org/0000-0003-2946-9497

 Michael Malim

Editor-in-Chief

PLOS Pathogens

orcid.org/0000-0002-7699-2064

**Additional Editor Comments :**

The authors need to clarify and correct inconsistencies in data presentation, video labeling, and interpretation, particularly regarding velocity comparisons, stimulus specificity, and the coherence of the mathematical model. They should also elaborate on the biological relevance of Capsaspora’s chemokinetic response, including comparisons between schistosome material and generic stimuli like BSA, and discuss potential in vivo implications. Additionally, the manuscript would benefit from clearer explanations of experimental design, key concepts such as aggregation versus chemokinesis, residence time, and collective behavior, as well as improved consistency in terminology, figure captions, and acronym usage.

**Journal Requirements:**

https://journals.plos.org/plospathogens/s/submission-guidelines#loc-parts-of-a-submission

- ® on pages: 17, and 18

- TM on page: 20.

5) We notice that your supplementary Figures, Table, and information are included in the manuscript file. Please remove them and upload them with the file type 'Supporting Information'. Please ensure that each Supporting Information file has a legend listed in the manuscript after the references list.

6) In the online submission form, you indicated that the other raw image stacks (~100 GB total), "can be provided upon request to the corresponding author (jpgerdt@iu.edu)." All PLOS journals now require all data underlying the findings described in their manuscript to be freely available to other researchers, either

1. In a public repository

2. Within the manuscript itself

3. Uploaded as supplementary information.

7) Please provide a completed 'Competing Interests' statement, including any COIs declared by your co-authors. If you have no competing interests to declare, please state "The authors have declared that no competing interests exist". Otherwise please declare all competing interests beginning with the statement "I have read the journal's policy and the authors of this manuscript have the following competing interests:"

**Reviewers' Comments:**

Reviewer's Responses to Questions

**Part I - Summary**

Reviewer #1: This paper is an account of the behavior of single-celled amoeboid predators Capsaspora owczarzaki and their motile response to chemoaffectors and chemical gradients. In particular, the discussion is focused on gradients associated with Schistosomes, on which they prey. This is a detailed piece of work that describes a chemokinetic effect (cells move faster in the presence of certain chemicals), and sets out to determine which chemicals elicit the strongest chemokinesis, and which have no effect. The results are used as input for mathematical modelling that predicts the usefulness of the chemokinetic/tactic effects in targeting prey cells.

The results show that the chemokinetic effect is fairly general and that Capsaspora shows a transient speed-up in the presence of a number of different macromolecular additives including bacterial and mammalian cell lysate as well as bovine and human serum albumins. This effect is subject to a saturation effect that is not restored by later addition of further stimulant.

In general, I think that this paper is a helpful addition to the literature, and gives new insight into an understudied species. It's unfortunate that no specific chemical signal triggering chemotaxis was found, but that is often the case in this type of experiment, and it's not a particular weakness - others could find the information useful.

Reviewer #2: Report of the manuscript entitled:” A close unicellular animal relative and predator of schistosomes exhibits chemokinesis in response to proteins and peptides from its prey” , authored by: Soniya R Quick et al.

This manuscript reports interesting observations on the chemokinesis behaviour of a unicellular close relative of animals, Capsaspora owczarzaki, which is also a potential predator of schistosomes. The authors investigate the nature of the substances behind the chemokinesis and chemotaxis behaviour in Capsaspora and also develop a mathematical model. The observations are interesting and this could be a valuable study for the communities of both, human pathogens and origin of multicellularity. Nevertheless, I had difficulties in understanding the path from the observations to the conclusions with the actual data. Some statements seem to me contradictory and also, I could not make sense of the mathematical model. See as follows my major and minor points.

Major

• Figure 1. Interestingly the mean value of velocity when exposed to schistosome (c), is around 6 microm/minute which is very similar to the mean velocity that appears to have the Capsaspora control in D/E/F and G. Does this mean that schistosomes are the less powerful trigger from all the ones assayed? Or mean velocities vary a lot from experiment to experiment and are not comparable? It would be nice that the authors elaborate a bit on this.

• Videos S1 and S3 correspond to Capsaspora with schistosome and S4 to S6 without schistosome, so reference in the text (figure Legend 1 and first paragraph of results) is wrong. Also difficult to make sense with the legend in the Fig Share where states that all videos had schistosome, but different treatment of the flasks, but in the manuscript only talks about the presence/absence of schistosome. Please unify and clarify. FIG SHARE: “ Videos of Capsaspora responding to schistosome sporocyst on plain tissue culture-treated plastic surface (Videos 1-3) and on fibronectin-treated surface (Videos 4-6).”

• The authors discover that FBS is also an inducer of chemokinesis in Capsaspora. Since this substance (at the same % that the authors use it) induces aggregation in Capsaspora, I wonder how the authors are able to induce separately or specifically one of the two behaviours in the cells. I have checked in materials and methods but neither there nor in the results there is an explanation in this regard. Because Capsaspora has been shown to be a model for aggregation behaviour and temporal multicellular structures, I think that would be important that this explanation is given, either in the results or in the M&M.

• Capsaspora chemokinesis is enhanced by lysates of unrelated types of cells, even the ones from the proven host Biomphalaria, which does not seem very specific and therefore maybe unrelated to a biological role of any of these lysates into the behaviour of Capsaspora. Can the authors speculate on the nature of the response from Capsaspora cells? Maybe something that all these lysates might have in common that is enough to trigger this response?

• Regarding the mathematical model there are some things that are not very clear to me. For example this statement “High chemo-effector levels lead to increased locomotion speed, which enhances diffusion due to random cell motion. This should not be a useful response for predators, as it would cause them to diffuse more rapidly as they approach their prey, transporting them away from their target” I understood innitially that the chemoeffector levels may difuse more due to increase cell movement, but in the second half of the sentence I understand that the cells are the ones difusing more rapidly. So, this is a bit confusing. Moreover a bit further down the statement “If the prey releases a large amount of chemokinesis-inducing chemicals, this would increase cell speed, and thus diffusion, encouraging higher dispersal away from the region where there is no longer prey ” seems to favour that this type of difusion would actually be advantatgeous for the predator since it would be driving away the predatory cells from those prey that were already lysated. But how this reconcilies with the observation that Capsapora rapidly prays on schistosomes if any of them has been lysated?

• Second statement that should be clarified :” However, coupled with irreversible attachment, the chemokinetically increased arrival rate at the schistosome implies a greater residence time for protists, which would be advantageous for predation. “ I don’t understand why the residence time increases in this circumstances, what it might increase is the number of predators arriving and attaching to the prey. Please clarify.

• Also, I understand that these are simulations, but I am surprised by the 15 minutes max. is that number representative of what happens in the culture? 15 minutes seems very fast to me.

For all of the above, for me it has been difficult to judge how much value and novelty the mathematical model adds to the study. Although the purpose and conclusions of the model are very well explained at the end of the introduction, I could not find in a understandable manner the proofs in the results to sustain it.

• For Capsapora aggregation it is known that calcium is also needed to trigger this specific behaviour. I know that the authors are not exploring the internal pathways by which this chemokinetic behaviour is possible, nevertheless it would be nice if they speculate taking into account what is known for aggregation in Capsaspora or even for the case of Dyctiostelium.

Minor

• In the abstract there is a sentence where there is a “it” that should be a “its”? “Until now, little was known of how Capsaspora regulates it rate and direction”

• Sentence with an extra “Also”: “We also have also not observed “

**Part II – Major Issues: Key Experiments Required for Acceptance**

Reviewer #1: No major additions in terms of experiments or simulations.

Reviewer #2: I am not sure if more experiments are needed, but the conclusions are not clear to me. My background does not allow me to revise the methodological details of the mathematical model but some of the defining premises seem award to me based on what is known of Capsaspora, including what is found in this study. As I mentioned above it could be because is not well explained.

**Part III – Minor Issues: Editorial and Data Presentation Modifications**

Reviewer #1: 1) I felt that some of the biological relevance was lost when it was noted that Capsaspora is attracted both to 'leaking' Schistosomes, various cell lysates and BSA. In the internal environment of a snail, there could presumably be any number of competing chemical gradients. Could the authors discuss any evidence that the attraction to material from schistosomes is specific/stronger than that to BSA, and whether future experiments to analyze the biological relevance of the effect might be worthwhile, and if so, which ones?

2) the abstract mentions that Capsaspora could be a model for the unicellular ancestor of animals. This aspect doesn't seem to be picked up in the discussion or conclusion; can the authors elaborate on this?

3) The collective behavior is interesting; can the authors confirm that the initial attachment of one Capsaspora cell is random? I think that they say this at some point but couldn't find mention in the section entitled "Chemokinesis response is cell-density-dependent but not contact-dependent", which seems like a relevant place.

4) There's a statement on p.17 of the review document (paragraph beginning "previous mathematical models [...]") that chemokinesis doesn't alter the trajectories, only the speed at which they're traversed (or words to that effect), which I agree with. I didn't quite understand the following statement that 'the number of cells that reach the schistosome are unchanged by the speed of locomotion', especially in light of the last sentence of that paragraph about arrival rates increasing in the case of irreversible attachment. Can the authors clarify?

5) Following from this, can the authors clarify what they mean by 'residence times' of the Capsaspora cells at the schistosome in the main document? It's discussed in the SI, but it's quite important to define it in the main body too. The same goes for the differences in the simulation of HSA and BSA, as it's not clear what these will mean in practice from the main text.

6) On a related note, can the authors clarify the following sentence: "We considered the possibility that the long-term attachment of Capsaspora to prey may be sufficient to make chemokinesis a beneficial trait; however, we found that to be untrue." Can they point to which experiment/simulation shows this, or was it not tested directly?

Typos/presentational issues:

a) Fig. 1B describes the mean distance migrated towards/away from a schistosome. Can the authors include in the caption how long the cells were monitored for? The details are probably in the methods somewhere, but the information is useful here.

b) Along similar lines, the caption to Fig. 1B mentions that only Capsaspora cells 70-240um from the schistosome were tracked. Can the authors explain why? This might be due to something straightforward, e.g. the microscope's fixed field of view.

c) 4-5 lines above Fig. 3: should this read "They slightly increased motility at a single [...]"? i.e. 'increased' rather than 'induced'?

d) Could the authors explain the acronym 'HSA' at first use in the main document?

Reviewer #2: There are formal mistakes in the figure legends of the Videos both in the text and in Fig share (see complete report) and a few typos (also mentioned in the full report).

PLOS authors have the option to publish the peer review history of their article (what does this mean? ). If published, this will include your full peer review and any attached files.

**Do you want your identity to be public for this peer review?** For information about this choice, including consent withdrawal, please see our Privacy Policy .

Reviewer #1: No

Reviewer #2: No

**Figure resubmission:**
---

## [Editor Report · Decision Letter 1]

9 Aug 2025

Dear Professor Gerdt,

We are pleased to inform you that your manuscript 'A close unicellular animal relative and predator of schistosomes exhibits chemokinesis in response to proteins and peptides from its prey' has been provisionally accepted for publication in PLOS Pathogens.

Best regards,

Dominique Soldati-Favre

Section Editor

PLOS Pathogens

Sumita Bhaduri-McIntosh

Editor-in-Chief

PLOS Pathogens

orcid.org/0000-0003-2946-9497

Michael Malim

Editor-in-Chief

PLOS Pathogens

orcid.org/0000-0002-7699-2064
---

## [Editor Report · Acceptance letter]

Dear Professor Gerdt,

We are delighted to inform you that your manuscript, " 

A close unicellular animal relative and predator of schistosomes exhibits chemokinesis in response to proteins and peptides from its prey," has been formally accepted for publication in PLOS Pathogens.

Best regards,

Sumita Bhaduri-McIntosh

Editor-in-Chief

PLOS Pathogens

orcid.org/0000-0003-2946-9497

Michael Malim

Editor-in-Chief

PLOS Pathogens

orcid.org/0000-0002-7699-2064